# Charity financial support to motor neuron disease (MND) patients in Greater London: the impact of patients' socioeconomic status—a cross-sectional study

Anna Gkiouleka,[1,2] Alison Manning,[3] Dianna Smith,[4] Andrea Malaspina,[5] Valentina Gallo[1,6,7]

For numbered affiliations see end of article.

**Correspondence to**
Dr Valentina Gallo;
v.gallo@qmul.ac.uk

## ABSTRACT

**Objective** There is an immense socioeconomic burden for both the patients with motor neuron disease (MND) and their families. The aim of this study is to evaluate the extent to which the provision offered by the Motor Neurone Disease Association is distributed among patients with MND living in the ethnically and socially diverse area of Greater London, according to the patients' socioeconomic situation and needs.

**Setting** Greater London, where age and sex-adjusted prevalence rates of MND in 2016 were calculated.

**Participants** Prevalent MND cases in Greater London, using anonymised data extracted from the Association's database.

**Exposure** Demographic and socioeconomic characteristics

**Primary and secondary outcome measures** Receiving a Motor Neurone Disease Association grant, and the amount of money received.

**Results** 396 individuals with amyotrophic lateral sclerosis were detected, the age-specific and sex-specific prevalence of MND was 4.02 per 100 000 inhabitants, higher among men (5.13 per 100 000) than women (3.01 per 100 000). Men were statistically significantly 40% less likely to receive a grant compared with women; among grant recipients, the younger the age of the participant, the higher the size of the grant received. The Index of Multiple Deprivation was not associated with either receiving a grant nor the amount of money received, among recipients.

**Conclusion** Financial support by the Motor Neurone Disease Association is provided across individuals and across boroughs regardless of their socioeconomic circumstances. Differences that benefits women and younger patients were detected.

## Strengths and limitations of this study

► Based on data available to the Motor Neurone Disease Association, the age-adjusted and sex-adjusted prevalence of motor neuron disease (MND) in Greater London could be calculated.

► Data available in the Motor Neurone Disease Association database cover the entirety of greater London allowing for comparison across boroughs and highly socioeconomically diverse areas.

► The analysis of socioeconomic status might be limited by the use of the index of multiple deprivation at area level as proxy.

► Information bias (more accurate information were collected for people who received a grant compared with those who did not) and selection bias (lack of information about people living with MND in Greater London, but not Motor Neurone Disease Association members) cannot be entirely ruled out.

## INTRODUCTION

The Motor Neurone Disease Association is a charity focused on motor neuron disease (MND) care, research and campaigning with approximately 8000 members in England, Wales and Northern Ireland. Its mission is to improve the care and the support for people with MND, their families and carers, to campaign and raise awareness about the needs of people living with MND and to fund and promote relevant research.[1] The current study is part of a broader research project initiated by the Association investigating the sociodemographic characteristics of patients with MND in Greater London, the extent of their met and unmet needs and their utilisation of services and resources offered by the Association.

Amyotrophic lateral sclerosis (ALS), the most common form of MND, is an idiopathic, fatal neurodegenerative disease.[2] Currently, it is estimated that there are approximately 5000 patients diagnosed with ALS in the UK, with an average age of onset of 55 years. The prevalence of ALS is 4–6 cases per 100 000 population, with a lifetime ALS risk ranging

from 1/600 to 1/1000.[1 3 4] Although diverse in its presentation, course and progression, ALS causes progressive muscle atrophy, which results in physical disabilities, loss of independence and eventually death due to respiratory muscle failure. Hence, the socioeconomic burden of the disease is immense both for the patients and their carers.[4]

Although the socioeconomic implications of ALS may be similar for all the patients (eg, income reduction, increased health expenses), their impact on the everyday reality of the patients and their families is subject to their socioeconomic status prior to the onset of the disease. Current evidence suggests that people belonging to the lower socioeconomic strata tend to suffer worse health and to have less access to healthcare and health-promoting resources.[5 6] In Britain, health inequalities persist as low income and low levels of subjective financial well-being are associated with poorer health in midlife and older age.[7] From this perspective, it is likely that patients with MND with lower socioeconomic status bear an increased health but also socioeconomic disadvantage. This implies that they have less means to maintain a human quality of life after the ALS onset but also that the socioeconomic consequences of the disease have a devastating impact on them and their close environment.

Moreover, studies on ethnic inequalities have revealed considerable gaps in self-rated health comparing ethnic minorities to the white British population. Analysis of 2011 census data has revealed that the White Gypsy or Irish Traveller men and women suffer a significant health disadvantage compared with the white British group, while the rest of the ethnic minorities report similar or better health outcomes than the white British group. This pattern appears stronger among men, as women in most cases either report worse health than the white British group (eg, Pakistani, Bangladeshi, Arab, Black Caribbean) or their health advantage is smaller than that of their male counterparts (eg, Black African, Indian).[5 7] At older ages, ethnic minorities suffer higher levels of morbidity compared with the white British population.[5 8] In London, the most ethnically diverse area of England and the region where the highest socioeconomic inequalities are observed, ethnic health inequalities are much more severe than in the rest of the country.[8 9] These findings suggest that beyond the pure economic, other dimensions of social positioning like ethnicity, gender or domicile have also a crucial impact on people's health and hence shape health inequalities among social groups. These inequalities translate both into an increased vulnerability to poor health, and into less capabilities to deal with disease.[10] People living with MND in London experience a unique situation of being part of a population structure diverse in terms of demographic characteristics, with a number of socioeconomic and health inequalities within groups.

The Motor Neurone Disease Association acknowledging both the financial implications of the disease for the patients' lives and the existing socioeconomic inequalities in Greater London concentrates its efforts in providing direct (ie, grants) or indirect (ie, equipment) financial support to patients with MND in the area. For this, all patients are informed by their care centre about the existence of the Association and those patients who express their interest in initiating a contact get referred to it by the care centre. Next, the Association sends to them extensive information material about the challenges of the disease, the available services offered by the National Health Service and those offered by the Motor Neurone Disease Association including financial support. Thus, patients can claim for it according to their needs. Their claims are usually made directly to the regional care development adviser in a personal manner and they get approved up to a maximum of 2000 pounds. The Motor Neurone Disease Association is the only large association working for people living with MND in London.

The overarching aim of this study was to evaluate if support given by the Motor Neurone Disease Association was distributed equally or fairly among people living with MND in London. As a consequence, we aimed to describe the provision offered by the Motor Neurone Disease Association among patients with MND living in the ethnically and socially diverse area of Greater London, according to the patients' demographic and socioeconomic situation and needs. The main objectives were first to estimate the prevalence of MND in London, starting from the data held by the Motor Neurone Disease Association, and, second, to explore whether the patients' socioeconomic position, gender and age were associated with the likelihood of receiving a grant, and with the amount of the grant received from the Motor Neurone Disease Association either in the form of cash or equipment (eg, riser recliners).

## METHODS
### Source of data
The Motor Neurone Disease Association database of people living with MND was accessed once the data had been anonymised. Data added to the data set before 2013 were partially incomplete; therefore, records added on 31 December 2012 or before and with a date of death before 1 January 2016 or missing were excluded from the analysis. Period prevalence of MND in London (January to December 2016) was computed including all people who were diagnosed with MND and alive for all or part of the year 2016. The Motor Neurone Disease Association expenditure for grant or equipment in the area of Greater London was collected through the Association's database that was updated with accurate information about financial grants and medical equipment supplied by the Association's staff for routing expenditure monitoring and then extracted for the purposes of this study. Information on gender was complete, while information on current age on 30 June 2016 was available for all but 71 patients. On the other hand, ethnicity and disease duration was mainly available for the grant receivers, implying a more consistent communication between the patients

and the Association. Each patient was georeferenced at Lower Super Output Area (LSOA 2015)[11] using their postcode. Each LSOA was used to approximate a neighbourhood that has a mean population of 1500.

Participant age was measured in years, disease duration was measured in months from the date of the diagnosis until 31 December 2016 or date of death, if antecedent. Individual level socioeconomic status was approximated using the 2015 Index of Multiple Deprivation (IMD)[12] at LSOA level and ethnicity was classified using the categories White, Black, Asian, Mixed ethnicity, and Other/unknown, according to the classification used in the 2011 Census. More detailed classification was avoided due to small numbers. The IMD ranks every small area in England from 1 (most deprived area) to 32 844 (least deprived area); this ranking is then divided in deciles which corresponds to most deprived 10%, 20%, etc. Financial support from the Motor Neurone Disease Association was measured as the total amount of money offered per patient either in the form of cash or medical equipment during the 1-year period. For the linear regression analysis, this value has been log transformed in order to meet the assumption of normality of distribution.

For the direct standardisation, age-specific and gender-specific London population by borough was extracted from the census data available from the National Online Manpower Information System (NOMIS), official labour market statistics.[13]

This research has been approved by the Queen Mary, University of London Ethical Committee on 26th of Semptember, 2016 (QMERC2016/40). No additional data are available for this analysis.

## Data analysis

Individuals with missing information on age (n=71) were recoded with the sex-specific and borough-specific median age category. A direct standardisation against WHO standard population separated for men and women was carried out in order to calculate age-adjusted and sex-adjusted prevalence rates.

A logistic regression to explore the socioeconomic determinants associated with receiving versus non-receiving a Motor Neurone Disease Association grant was conducted. Among people receiving some form of grants, a linear regression exploring the association of the same determinants and the amount of money awarded to recipients was subsequently run.

Non-linearity of the association between grant amount and age has been tested adding the age-squared term. In addition, a quintile analysis of the grant amount has been conducted to assess if any specific group is more likely to receive exceptionally large amount of money.

A descriptive geographical analysis was conducted using ArcGIS V.10.4. Data on age-standardised prevalence of MND were mapped at borough level, due to small patient numbers. Additional mapping of the funding from grants or equipment expenditure per patient was completed at the same scale to identify geographic variation in provision. These data were mapped against the median borough level of the IMD. Statistical analysis was conducted using the STATA V.12.1 package.

## Patient and public involvement

There was no patient nor public involvement in the study design or analysis. Results will be made available to patients through the Motor Neurone Disease Association website.

## RESULTS

A total of 983 observations were extracted from the Motor Neurone Disease Association data set, after removal of four duplicates and one observation with missing value for all dates. Of these, 569 observations were removed because the inclusion criteria for the period prevalence of being alive for all of part of 2016 could not be verified as they were added on or before 31 December 2012 but had a missing date of death. Further, 18 records were deleted as they were added on the data set on or after 1 January 2017 and no date of diagnosis was recorded. The final analysis was conducted on a total of 396 prevalent MND cases (233 men and 163 women) in Greater London in 2016.

Direct standardisation was conducted on the total sample of 396 individuals in order to calculate the age-specific and sex-specific prevalence, by borough. In London, globally, the age-standardised prevalence of MND was 4.02 per 100 000 inhabitants; this was higher among men (5.13 per 100 000) than women (3.01 per 100 000). Standardised sex-specific and borough-specific prevalence per 100 000 is shown in figure 1. In table 1, the population of London, number of people living with MND and their mean and median age by gender are reported alongside the directly standardised prevalence rates per 100 000 inhabitants by borough.

Demographic characteristics of people living with MND in London are reported in table 2. Men were more prevalent than women (58.8%), and the overall mean age was 65.7 years (SD 12.7). Whites are the largest ethnic group (32.1%) followed by Asians (6.3%) and Blacks (3.3%). The mean decile of the IMD is 5.4 (SD 2.7), close to the median (table 2).

The distribution of grant awarding was found to vary across ethnic groups with higher proportions of Black (38.5%) and Asian (48.0%) people receiving a grant compared with whites (32.3%) (table 2). However, as shown by boxplots in figure 2, among white patients, there was the greatest variability in amounts of financial support received with a few cases having received much larger amounts than the average.

In figure 3, the proportion of people living with MND who have received a grant for the Motor Neurone Disease Association, by borough, is plotted against the borough-specific IMD quantiles. At least one in four people received economic support in four out of six

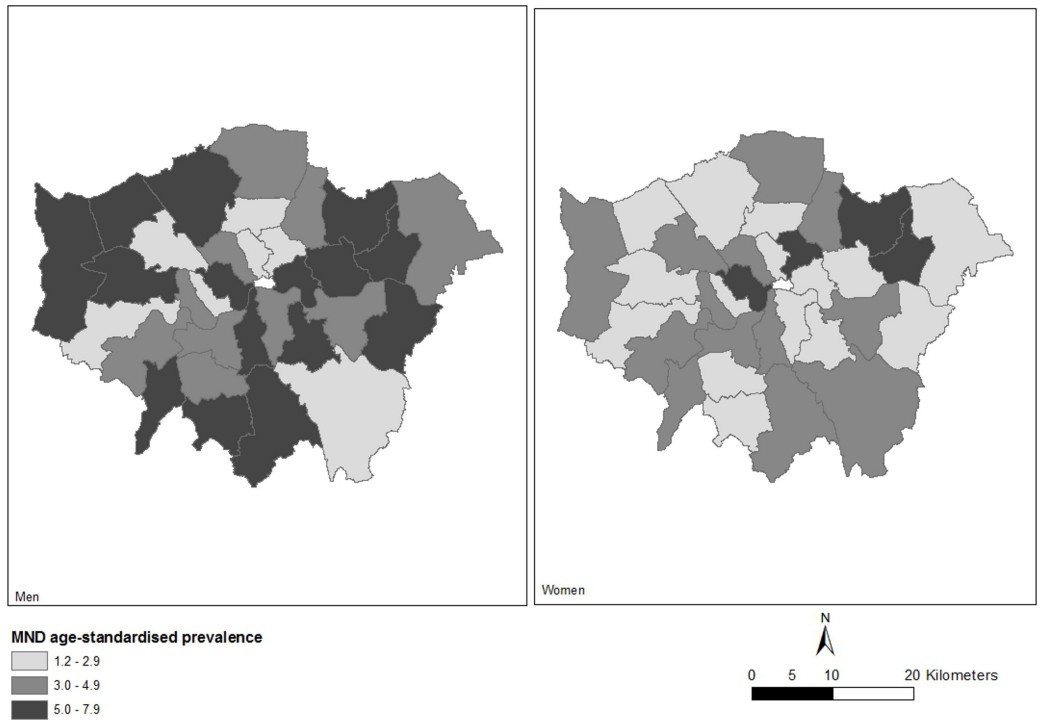

**MND age-standardised prevalence**
- 1.2 - 2.9
- 3.0 - 4.9
- 5.0 - 7.9

**Figure 1** Map of Greater London showing the prevalence of motor neuron disease (MND) by borough and gender.

least deprived boroughs, and in only two out of six most deprived boroughs.

Results of the logistic regression investigating factors associated with the Motor Neurone Disease Association grant award are shown in table 3. Only belonging to an 'other' or unknown ethnic group was inversely associated with receiving a grant in the crude analysis. In the adjusted model, men were statistically significantly 40% less likely to receive a grant compared with women; people of other/unknown ethnic background were also 60% less likely to receive a grant compared with whites. When investigating determinants of the amount of money granted, among grant recipients only, age was the only factor found to be negatively significantly associated with amount of grant received: the younger the age of the participant, the higher the size of the grant received (table 3). Adding age as a quadratic term did not show a better fit compared with the linear association; a quintile analysis for IMD yield to very similar results (results now shown). Interestingly, IMD decile, used as a proxy for socioeconomic status of the participants, was not associated with either receiving a grant nor the amount of money received, among recipients.

## DISCUSSION

The analysis of financial support offered by the Motor Neurone Disease Association to people living with MND in Greater London showed that women were more likely than men to receive a grant, and—among grant recipients—younger people receive more money than older ones. The finding that people from other/unknown ethnic groups are less likely than whites to receive a

grant is likely due an information bias; in fact, records of people receiving grants are more likely to be accurate. Importantly, no significant difference was seen among ethnic groups nor across socioeconomic strata. However, these findings need to be interpreted with caution as no information was available on people living with MND in London who were not members of the Motor Neurone Disease Association. The potential selection bias underlying this could have biased towards the null a possible positive association between receiving a grant and socioeconomic position.

### Gender inequalities

Women's greater probability to receive a grant compared with men could be explained on the basis of culturally dominant constructions of masculinity and femininity that affect men's and women's help seeking behaviours.[14] From this perspective, men, in order to be consistent with the dominant masculine representation, tend to be less eager to seek for medical or social support as such behaviour is associated with weakness and vulnerability, attributes that are traditionally associated with women.[15] In this light, our findings might suggest that men, due to their greater reluctance to seek help, are also less willing to seek for financial support provided by the Motor Neurone Disease Association. Moreover, due to these gender stereotypes, women in general are also responsible for providing care to the members of their family. This might mean that they are more familiar with healthcare and social support institutions and hence are better informed and skilled in navigating the services. Therefore, the greater probability to receive financial support from the Motor Neurone Disease Association might be derived

**Table 1** Total population, number of people living with motor neuron disease (MND), mean, median age and standardised prevalence per 100 000 by gender and borough in Greater London in 2016

| Borough | Total population | People with MND | Men Mean/median age | Men Standardised prevalence per 100 000 | Women Mean/median age | Women Standardised prevalence per 100 000 |
|---|---|---|---|---|---|---|
| Barking and Dagenham | 185 911 | 12 | 64/61 | 6.06 | 66/67 | 4.90 |
| Barnet | 356 386 | 22 | 64/66 | 7.04 | 77/74 | 2.16 |
| Bexley | 231 997 | 14 | 70/69 | 5.76 | 63/63 | 2.14 |
| Brent | 311 215 | <10 | 43/43 | 1.21 | 67/67 | 3.06 |
| Bromley | 309 392 | 23 | 63/65 | 2.33 | 70/68 | 4.89 |
| Camden | 220 338 | 13 | 62/65 | 3.41 | 52/60 | 4.07 |
| Croydon | 363 378 | 22 | 72/68 | 5.48 | 70/71 | 3.52 |
| Ealing | 338 449 | 15 | 53/52 | 6.08 | 75/72 | 1.69 |
| Enfield | 312 466 | 18 | 63/66 | 4.11 | 63/63 | 4.18 |
| Greenwich | 254 557 | 13 | 63/66 | 4.30 | 63/63 | 3.28 |
| Hackney | 246 270 | <10 | 63/63 | 2.17 | 64/65 | 6.80 |
| Hammersmith and Fulham | 182 493 | <10 | 68/68 | 3.71 | 60/60 | 3.25 |
| Haringey | 254 926 | <10 | 59/59 | 1.74 | 40/40 | 0.65 |
| Harrow | 239 056 | 15 | 66/66 | 7.14 | 76/76 | 1.78 |
| Havering | 237 232 | 15 | 71/72 | 4.76 | 69/69 | 2.29 |
| Hillingdon | 273 936 | 14 | 69/71 | 5.03 | 57/54 | 2.93 |
| Hounslow | 253 957 | <10 | 62/62 | 2.80 | 50/43 | 2.80 |
| Islington | 206 125 | <10 | 78/78 | 1.97 | | 0.00 |
| Kensington and Chelsea | 158 649 | <10 | 70/67 | 2.29 | 87/86 | 2.38 |
| Kingston on Thames | 160 060 | 11 | 61/62 | 6.93 | 68/69 | 3.59 |
| Lambeth | 303 086 | 11 | 61/60 | 5.23 | 63/62 | 3.10 |
| Lewisham | 275 885 | 12 | 57/54 | 5.54 | 65/60 | 2.36 |
| Merton | 199 693 | <10 | 67/73 | 4.77 | 63/51 | 2.06 |
| Newham | 307 984 | 11 | 60/60 | 7.93 | 63/63 | 0.82 |
| Redbridge | 278 970 | 24 | 67/66 | 7.93 | 74/79 | 4.93 |
| Richmond on Thames | 186 990 | 11 | 72/74 | 3.13 | 71/72 | 4.36 |
| Southwark | 288 283 | <10 | 65/70 | 4.13 | 79/79 | 1.18 |
| Sutton | 190 146 | 12 | 62/63 | 6.27 | 68/70 | 2.87 |
| Tower Hamlets | 254 096 | <10 | 61/59 | 7.36 | 48/48 | 1.15 |
| Waltham Forest | 258 249 | 11 | 67/70 | 4.14 | 71/72 | 3.00 |
| Wandsworth | 306 995 | 13 | 72/73 | 4.36 | 69/74 | 3.80 |
| Westminster | 219 396 | 14 | 60/60 | 4.92 | 66/67 | 5.95 |
| Total | 8 166 566 | 396 | 64 (SD 12)/ 65 (IQR 57–72) | 5.13 | 68 (SD 13) 70 (IQR 61–76) | 3.01 |

from the fact that women are more informed about the support opportunities or they simply know better where to look for such opportunities compared with men.

On the other hand, another possible interpretation of this association is the presence of a greater need among women compared with men. Given that women are in general more likely to be unemployed, to have more and longer gaps in their working history (eg, due to pregnancy) or to work part-time and in less regulated sectors than men implies that they also have less access to social provision.[16] So, a greater utilisation of services provided from non-state institutions (ie, those provided by charities such as the Motor Neurone Disease Association) might shed light on a gap in social provision that women seek to cover via alternative sources. Future research could reveal which of these explanations apply to patients with MND or whether they all apply at the same time. In any case, the findings suggest that the utilisation of the

**Table 2** General characteristics of the population living with MND in greater London, and divided by being recipient of a grant from the Motor Neurone Disease Association

| Characteristics | All n=396 | Grant recipient n=96 | Not grant recipient n=300 | P value* |
|---|---|---|---|---|
| Men (%) | 233 (58.8) | 49 (51.0) | 184 (61.3) | 0.075 |
| Age, mean (SD) | 65.7 (12.7) | 65.0 (13.0) | 66.0 (12.6) | 0.523 |
| Ethnicity | | | | |
| White (%) | 127 (32.1) | 41 (32.3) | 86 (67.7) | <0.001 |
| Asian (%) | 25 (6.3) | 12 (48.0) | 13 (52.0) | |
| Black (%) | 13 (3.3) | 5 (38.5) | 8 (61.5) | |
| Mixed | 7 (1.8) | 2 (28.6) | 5 (71.4) | |
| Other/unknown | 224 (56.6) | 188 (83.9) | 36 (16.1) | |
| Index of Multiple Deprivation | | | | |
| Decile, mean (SD) | 5.4 (2.7) | 5.5 (2.7) | 5.4 (2.6) | 0.773 |

*P value from $\chi^2$ test for categorical variables, and t-test for continuous variables.
MND, motor neuron disease.

Motor Neurone Disease Association's financial services is subject to gender differences and this should be taken into consideration in the future communication strategy of the Association.

However, a cautious interpretation of this findings is needed, as we cannot exclude that men are over-represented in this sample due to a selection bias. This would have biased the observed association which would look stronger than is in reality.

### Age inequalities

The fact that older patients receive smaller amounts of grants from the Motor Neurone Disease Association might be due to the broader social marginalisation of older people. Older people tend to be less able to maintain social relationships within their communities and this makes them less able to participate in daily activities.[17] They often reside in spatially segregated areas

where public and private institutions are also absent.[18] Such circumstances may inhibit older patients with MND from seeking and receiving assistance from the Motor Neurone Disease Association—among other providers—especially when they also lack strong bonds with family or friends. Again, it is not possible to rule out that these findings could at least be in part due to selection bias in case older people were more likely to be over-represented in the observed sample.

### Social and ethnic inequalities

The framework of social determinants of health[19] suggests that socioeconomic status is not only an indicator of the actual material and social resources that one has but also of the ability a person has to make use of these resources to achieve their health potential.[20] In this frame, we would expect that the likelihood of patients to receive financial support from the Motor Neurone Disease Association

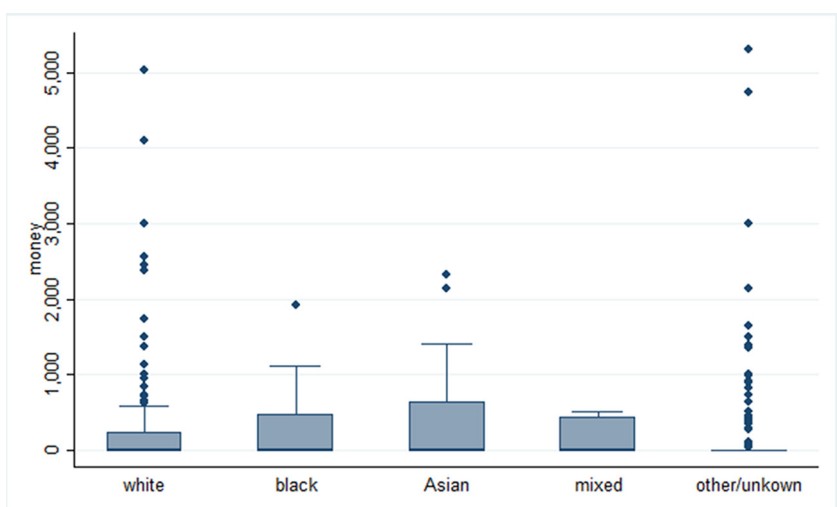

**Figure 2** Boxplots representing total spent in Great Britain Pound sterling (GBP) by the Motor Neurone Disease Association by ethnic group on Greater London (n=396).

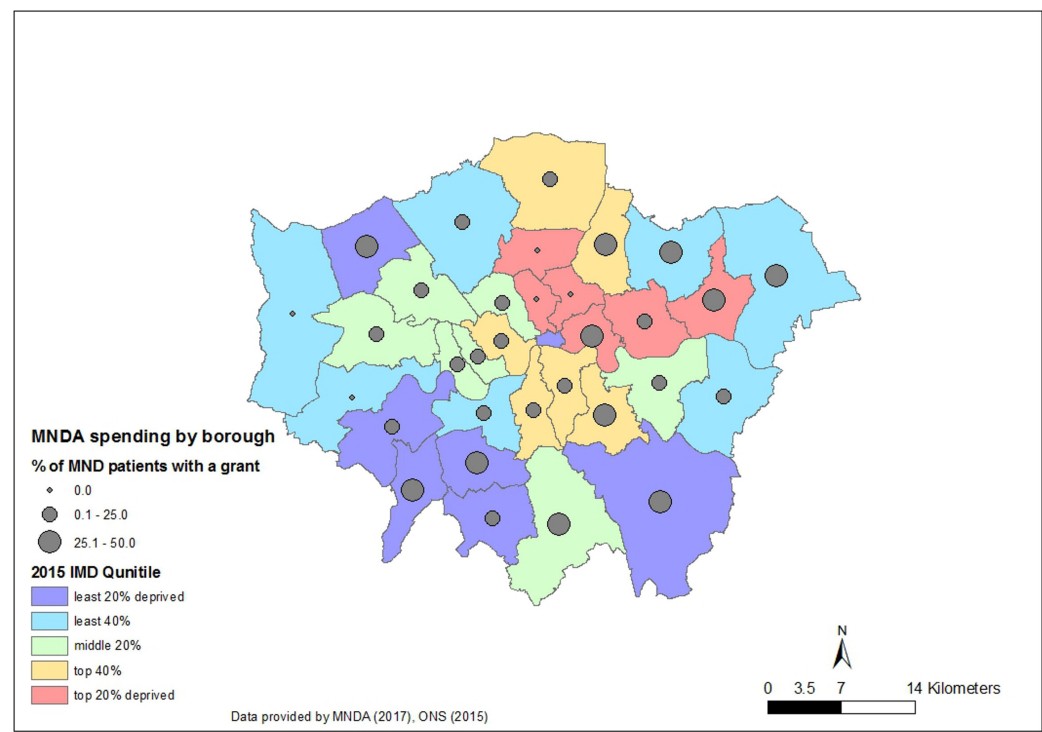

**Figure 3** Map of Greater London showing proportion of people living with motor neuron disease (MND) who received a grant from the Motor Neurone Disease Association, by borough, over the Index of Multiple Deprivation (IMD) quintile.

as well as the amount of their grant would be subject to their own socioeconomic status as an indicator of their everyday circumstances as well as of their capacity to deal with the challenges imposed by the disease. People in lower social strata not only have greater socioeconomic needs but have also less capacity to make the ultimate use of the available resources.[10] Hence, in pursuing a fair provision of financial support, we would expect the Motor Neurone

Disease Association to offer grants more often and more generously to patients with a lower socioeconomic status. However, finding no evidence for socioeconomic and ethnic inequalities in receiving financial support from the Motor Neurone Disease Association indicates a strategy of 'equal' treatment for all the patients from the side of the Association regardless of the patients' circumstances. However, our estimates could be biased towards the null if we assume the

**Table 3** OR and relative 95% CI coming from logistic regression models investigating determinants of being recipient of an MND grant among prevalent cases of MND in Greater London and Beta-coefficients and relative 95% CI from linear regression models investigating factors associated with amount of money awarded (iin GBP, log-transformed)

| | All prevalent cases n=396 | | Grant recipients only n=96 | |
|---|---|---|---|---|
| | Logistic regression Crude OR (95% CI) | Logistic regression Adjusted OR (95% CI) | Linear regression Crude β coefficient (95% CI) | Linear regression Adjusted β coefficient (95% CI) |
| Male gender | 0.66 (0.41 to 1.04) | 0.60 (0.37 to 0.98) | −0.078 (-0.58 to 0.42) | −0.11 (-0.62 to 0.39) |
| Age (years) | 0.99 (0.98 to 1.01) | 0.99 (0.98 to 1.01) | −0.020 (-0.04 to 0.001) | −0.02 (-0.04 to 0.002) |
| Ethnic group | | | | |
| White | Ref. | Ref. | Ref. | Ref. |
| Black | 1.31 (0.40 to 4.26) | 1.32 (0.39 to 4.48) | 0.52 (-0.65 to 1.69) | 0.33 (-0.86 to 1.53) |
| Asian | 1.94 (0.81 to 4.61) | 2.18 (0.89 to 5.32) | −0.08 (-0.89 to 0.73) | −0.26 (-1.09 to 0.57) |
| Mixed | 0.84 (0.16 to 4.51) | 0.82 (0.15 to 4.49) | −0.09 (-1.88 to 1.69) | −0.30 (-2.09 to 1.49) |
| Other/unknown | 0.40 (0.24 to 0.67) | 0.40 (0.24 to 0.68) | 0.17 (-0.39 to 0.73) | 0.10 (-0.46 to 0.67) |
| IMD (deciles) | 1.01 (0.93 to 1.10) | 1.04 (0.95 to 1.14) | −0.005 (-0.097 to 0.087) | 0.02 (-0.07 to 0.12) |

'Beta coefficients and relative 95% CI from linear regression models investigating factors associated with the amount of money awarded (in GBP, log transformed).' and 'MND, motor neuron disease'

presence of a selection bias making more wealthy people more likely to access the Motor Neurone Disease Association (and therefore being included in the present study) compared with more deprived people. Future research with more accurate data in terms of ethnicity and an individual measure of socioeconomic status would be useful in order to verify this pattern but also to reveal whether this 'equal treatment' approach is the most effective one for the specific population. Equal distribution of resources is not enough for dealing with people who stand in unequal social positions. This might be an issue for consideration for the Association's strategy for providing not only equal but also fair service. Also, a population-based registry would reduce the risk of selection bias and allow more precise estimations.

## Limitations

Considering the methods used for this analysis, and specifically the exclusion of observations with no recorded date of death, it is possible that this apparent lack of association is at least partially due to selection bias. If people from lower socioeconomic backgrounds and ethnic minorities are more likely to fail to remain engaged with the Association, they are also more likely to have missing data and therefore are being under-represented in the prevalence exercise. This would imply that prevalence among these subgroups would be artificially inflated. Moreover, it is difficult to estimate generalisability of these results, as data on prevalent MND cases that are not members of the Motor Neurone Disease Association are lacking. This could be a further source of selection bias as those people are more likely to be less educated or migrants. The lack of association between IMD and awards may be due to the scale of the analysis and the required assumption that patients in this study had the same socioeconomic status as the local population, as IMD is assigned by LSOA (mean population of 1500). This assumption is potentially subject to ecological fallacy,[21] as individual socioeconomic status cannot be always estimated accurately from IMD.[22] Moreover, the limited sample size of the analysis among grant recipients does not allow detection of smaller effects, which could be only achieved by increasing the power. Finally, although all the patients who get referred to the Motor Neurone Disease Association receive the information material about the available financial support, assuming that all of them read the material and are equally able to make contact with the Association in order to claim financial support is unlikely. So, future research should also explore the extent to which the information about the services available by the Association is distributed equally and whether there are certain patients who are less likely to be aware of their entitlements.

## CONCLUSIONS

The present results show that provision of financial support by the Motor Neurone Disease Association is provided across individuals and across boroughs regardless of their socioeconomic circumstances. However, it seems that there is a difference across gender and age that benefits women and younger patients, respectively. Ethnic differences have not been observed. Future research with a larger number of cases, spread across multiple regions, and more accurate data on the patients' socioeconomic position and ethnic background would allow us to confirm these findings and draw more certain conclusions about the effectiveness of the Motor Neurone Disease Association's strategy in distributing financial support among patients fairly.

**Author affiliations**
¹Centre for Primary Care and Public Health, Queen Mary University of London, London, UK
²Department of Sociology, University of York, York, UK
³Motor Neurone Disease Association, Northampton, UK
⁴Geography and Environmental Sciences, University of Southampton, Southampton, UK
⁵Department of Neuroscience and Trauma, Blizard Institute, Queen Mary University of London, London, UK
⁶Epidemiology and Medical Statistic Unit, London School of Hygiene and Tropical Medicine, London, UK
⁷School of Public Health, Imperial College London, London, UK

**Acknowledgements** This research has been made possible, thanks to a grant of the Motor Neurone Disease (MND) Association to VG (grant number GPPG1G5R). VG had full access to all of the data in the study and takes responsibility for the integrity of the data and the accuracy of the data analysis. She declares that this manuscript is honest, accurate and transparent account of the study being reported; that no important aspects of the study have been omitted. All co-authors had full access to the data (including statistical reports and tables), and can take responsibility for the integrity of the data and the accuracy of the data analysis.

**Contributors** Study concept and design: VG. Analysis and interpretation of data: VG, AG, DS. Drafting of the manuscript: AG. Data collection: AMan. Critical revision of the manuscript for important intellectual content: AMal, AMan, DS, VG.

**Funding** This article is part of the project 'Social inequalities in accessing services provided by the MND Association' funded by the MND Association (GPPG1G5R).

**Competing interests** None declared.

**Patient consent** Not required.

**Ethics approval** Queen Mary University of London Ethical Committee (QMERC2016/40).

**Provenance and peer review** Not commissioned; externally peer reviewed.

**Data sharing statement** No additional unpublished data is available for this study.

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
