## [Reviewer comments · BMJ Open]

ARTICLE DETAILS

TITLE (PROVISIONAL)	Charity financial support to Motor Neurone Disease (MND) patients in Greater London: the impact of patients' socio-economic status – a cross-sectional study
AUTHORS	Gkiouleka, Anna; Manning, Alison; Smith, Dianna; Malaspina, Andrea; Gallo, Valentina

VERSION 1 – REVIEW

REVIEWER	Lisa Graves Western Michigan University Homer Stryker M.D. School of Medicine, United States
REVIEW RETURNED	19-Apr-2018

GENERAL COMMENTS	This paper explores an interesting and often asked question in clinical practice. Common assumptions were questioned making it an interesting read. Sufficient context was provided for this reader to understand the grants system, but if the authors would include some explanation of the system it would be useful for international readers. Paper would benefit from a further description of the database and how accurate this is. Limitations section is well written, but this reader would have appreciated further discussion of selection bias and to see it threaded through the discussion.
---

REVIEWER	Job Harms Erasmus University, School of Economics, Groningen University, Faculty of Behavioral and Social Sciences (per May 1st)
REVIEW RETURNED	21-Apr-2018

GENERAL COMMENTS	This paper presents a study about the Motor Neuron Disease Association's allocation of support towards patients in the Greater London area. In particular, the study addresses how patients' demographics and socio-economic characteristics correlate with access to MNDA grants. The results indicate that males are more likely to receive grants, and that younger patients receive larger grants. Furthermore subjects socio-economic status, as proxied by the score of their residential neighborhood on the "Index of Multiple Deprivation", did not predict the likelihood of receiving a grant or the amount of grant money received by patients. These results suggest the association's support might not be targeted to the groups that are most vulnerable to MND, namely groups of lower socio-economic backgrounds. This is an interesting result and could provide both MNDA and other charities guidance as to how their resources might be more "fairly" allocated.
---

	Having said this, there are several points which could benefit from further explanation and analysis.  - First, the authors use a geographical proxy for socio-economic status, the IMD. However, this measure is somewhat noisy, particularly in postcodes with greater levels of inequality. This could explain why the IMD doesn't predict probability of receiving a grant. More precise estimates could be derived from using administrative micro-level socio-economic indicators, such as patients' household income, educational background etc. I don't know the English context, but wouldn't it be possible to request such microlevel data from the bureau of statistics? - Second, it remains unclear to the reader whether patients in the London area that suffer from NMD might also (can) receive support through other organizations/foundations. This will be important for the interpretation of the results. For example, if low SES patient groups already receive financial support through other programs it's not evident that the current allocation mechanism used by the MNDA is "unfair" as such. - Third, it remains unclear to the reader how grants are requested/allocated. Do patients have to submit a grant request with the association? If so, the results should be adjusted and interpreted accordingly: the association between patient characteristics and the probability of receiving a grant will be driven by the propensity of patients to request a grant in the first place. In this case it would make sense to run a logit/probit analysis where the dependent variable is whether or not a patient requests a grant. - Fourth, the sample size for the analysis on grant recipients (N=96) is rather limited. It would be valuable to conduct a power analysis to place the non-significant association between socio-economic status and grant amounts into perspective. Next, given the variability in grant amounts it might be interesting to run additional analyses, e.g. quantile regression, for example to see if certain SE groups are more likely to received very large grants. - Fifth, regarding the association between age and grant amounts received, it might be interesting to test for non-linearities, e.g. by including/adding "log age" or "age sq." terms in the model specification. - Sixth, it is unclear how the gender variable is coded. Is male=1 and female=2? Now the regression table 3 is a bit confusing, it shows a positive coefficient on the "male" variable yet the results section indicates that females are more likely to receive grants. This should be clarified. - Finally, while the research question is outlined in the last paragraph on p6, it would be useful to rephrase it somewhat to make it more clear and precise, ideally framed as an hypothesis. For example, you could apply the following wording for the hypothesis: "Patients' likelihood of receiving funding from MNDA is not predicted by socio-economic status". In addition, it would be useful to add a section where you explain why you formulate this hypothesis – or make explicit that the study is of an explorative nature and you don't have priors.
--	---

VERSION 1 – AUTHOR RESPONSE

Reviewer: 1

Reviewer Name: Lisa Graves

Dear Dr Graves,

Thank you very much for highlighting your interest in our paper and for your detailed comments. Please, find below our answers to your comments point by point.

Sufficient context was provided for this reader to understand the grants system, but if the authors would include some explanation of the system it would be useful for international readers.

This information is now available in a short paragraph in the introduction (see: page 5, paragraph 2) and its implications are also mentioned in the limitations section (see: page 11).

Paper would benefit from a further description of the database and how accurate this is.

This is now added (see Source of data section, page 7).

Limitations section is well written, but this reader would have appreciated further discussion of selection bias and to see it threaded through the discussion.

This is now discussed more in depth in the discussion section, and specifically in each sub-section (see: page 10, 11 and 12).

Reviewer: 2

Reviewer Name: Job Harms

Dear Dr Harms,

Thank you for your providing us with your constructive feedback. Your comments have helped us improving the quality of our paper. Please, read below our answer to the points you have stressed.

First, the authors use a geographical proxy for socio-economic status, the IMD. However, this measure is somewhat noisy, particularly in postcodes with greater levels of inequality. This could explain why the IMD doesn't predict probability of receiving a grant. More precise estimates could be derived from using administrative micro-level socio-economic indicators, such as patients' household income, educational background etc. I don't know the English context, but wouldn't it be possible to request such microlevel data from the bureau of statistics?

We agree that the IMD as a proxy of socioeconomic background is a noisy measure and that the results may appear different with a more accurate measure. However, our analysis is based on data provided exclusively by the MND Association databases. Unfortunately, no information about household or individual income was available and this is why we have finally used the IMD to run our analysis. However, we have made this point more explicit in the limitation

Second, it remains unclear to the reader whether patients in the London area that suffer from NMD might also (can) receive support through other organizations/foundations. This will be important for the interpretation of the results. For example, if low SES patient groups already receive financial

support through other programs it's not evident that the current allocation mechanism used by the MND Association is "unfair" as such.

In UK, all residents have access to the same services provided by the NHS, however, to what extent everyone has the same ability/motivation to access them is unclear. The MND Association assist people living with MND to access the services provided by the NHS, and – in addition – provides grants to cover expenses which are not covered by the NHS and/or social services (e.g. a wet room). MND Association is the only association working with people living with MND, therefore no other source of funding is readily available, unless people have other specific individual sources. We have made this explicit in the introduction (p. 6)

Third, it remains unclear to the reader how grants are requested/allocated. Do patients have to submit a grant request with the association? If so, the results should be adjusted and interpreted accordingly: the association between patient characteristics and the probability of receiving a grant will be driven by the propensity of patients to request a grant in the first place. In this case it would make sense to run a logit/probit analysis where the dependent variable is whether or not a patient requests a grant.

This information is now provided in page 6. Given that all claims get approved up to a given maximum, the probability of receiving a grant in fact is equivalent of the probability of asking for a grant.

Fourth, the sample size for the analysis on grant recipients (N=96) is rather limited. It would be valuable to conduct a power analysis to place the non-significant association between socio-economic status and grant amounts into perspective. Next, given the variability in grant amounts it might be interesting to run additional analyses, e.g. quantile regression, for example to see if certain SE groups are more likely to received very large grants.

We agree with your comment and thus, we now state explicitly that our small sample size does not allow us to detect smaller effects (see limitations section, page 11-12). Also, in order to deal with the variability in grants amounts, we ran our analysis using the log of the grant amount. After your suggestion, we also ran a quintile regression with the actual grant amount and no substantial differences were observed from our original models.

Fifth, regarding the association between age and grant amounts received, it might be interesting to test for non-linearities, e.g. by including/adding "log age" or "age sq." terms in the model specification.

We tested for the curvilinear effect of age by adding age squared in our analysis, but the age squared was not significant.

Sixth, it is unclear how the gender variable is coded. Is male=1 and female=2? Now the regression table 3 is a bit confusing, it shows a positive coefficient on the "male" variable yet the results section indicates that females are more likely to receive grants. This should be clarified.

We confirm that we coded female=0 and male=1, so the results in table should be interpreted comparing male to female. We have changes the table to reflect this.

Finally, while the research question is outlined in the last paragraph on p6, it would be useful to rephrase it somewhat to make it more clear and precise, ideally framed as an hypothesis. For example, you could apply the following wording for the hypothesis: "Patients' likelihood of receiving funding from MND is not predicted by socio-economic status". In addition, it would be useful to add a section where you explain why you formulate this hypothesis – or make explicit that the study is of an explorative nature and you don't have priors.

This is now changed. See page 7.

VERSION 2 – REVIEW

REVIEWER	Lisa Graves Western Michigan University Homer Stryker M.D. School of Medicine, U.S.A.
REVIEW RETURNED	05-Aug-2018
GENERAL COMMENTS	Thank you for the opportunity to review this revised manuscript. The revisions have addressed previous concerns. Importantly there is additional information explaining details of the grant process that make the paper more understandable for readers outside of the U.K. A more detailed description of the limitations of the paper as well as caution around the interpretation make this paper an interesting read.